# ConF: A Deep Learning Model Based on BiLSTM, CNN, and Cross Multi-Head Attention Mechanism for Noncoding RNA Family Prediction

**DOI:** 10.3390/biom13111643

**Published:** 2023-11-13

**Authors:** Shoryu Teragawa, Lei Wang

**Affiliations:** School of Software, Dalian University of Technology, Dalian 116024, China; lei.wang@dlut.edu.cn

**Keywords:** noncoding RNA, deep learning, gene expression

## Abstract

This paper presents ConF, a novel deep learning model designed for accurate and efficient prediction of noncoding RNA families. NcRNAs are essential functional RNA molecules involved in various cellular processes, including replication, transcription, and gene expression. Identifying ncRNA families is crucial for comprehensive RNA research, as ncRNAs within the same family often exhibit similar functionalities. Traditional experimental methods for identifying ncRNA families are time-consuming and labor-intensive. Computational approaches relying on annotated secondary structure data face limitations in handling complex structures like pseudoknots and have restricted applicability, resulting in suboptimal prediction performance. To overcome these challenges, ConF integrates mainstream techniques such as residual networks with dilated convolutions and cross multi-head attention mechanisms. By employing a combination of dual-layer convolutional networks and BiLSTM, ConF effectively captures intricate features embedded within RNA sequences. This feature extraction process leads to significantly improved prediction accuracy compared to existing methods. Experimental evaluations conducted using a single, publicly available dataset and applying ten-fold cross-validation demonstrate the superiority of ConF in terms of accuracy, sensitivity, and other performance metrics. Overall, ConF represents a promising solution for accurate and efficient ncRNA family prediction, addressing the limitations of traditional experimental and computational methods.

## 1. Introduction

RNA is a biopolymer composed of four nucleotides: adenine (A), uracil (U), guanine (G), and cytosine (C) [1]. Functionally, RNA can be categorized into *coding RNA* and *noncoding RNA* (*ncRNA*). While *ncRNAs* are derived from *ncRNA genes*, they do not encode proteins [2]. Nevertheless, they play significant roles in various cellular processes [3] and diseases [4] through mechanisms such as replication, transcription, and gene expression [5,6]. Extensive transcriptomics and bioinformatics studies have identified thousands of *ncRNAs* in humans, classified based on their functionality and length. Examples of *ncRNA* categories include *microRNA*, *ribosomal RNA* (*rRNA)*, *ribozymes*, *small nuclear RNA* (*snRNA*) including small *nucleolar RNA* (*snoRNA*), *transfer RNA* (*tRNA*), *intron RNA*, *internal ribosome entry site* (*IRES*), *leader*, and *riboswitch*. These *ncRNAs* exert crucial functions in organisms. For instance, *snRNA* processes *heteronuclear RNA* within the cell nucleus, regulates transcription factors, and maintains telomeres [7]. *Ribozymes*, serving as RNA enzymes in organs, facilitate the connection of amino acids during protein synthesis. tRNA acts as a physical bridge between *messenger RNA* (*mRNA*) and amino acid sequences [8]. *Intron RNA*, transcribed from intron genes, engages in extensive internal interactions post RNA transcription and aids in the proper ordering of exons [9,10]. *IRES* facilitates the binding of the ribosome to *mRNA*, initiating protein translation and synthesis [11]. The *leader* represents the upstream portion of the start codon in *mRNA* and assumes an important role in regulating *mRNA* transcription [12]. Riboswitches are regulatory segments within *mRNA* that can adopt specific conformations to modulate *mRNA* transcription processes [13]. Consequently, *ncRNAs* hold a critical position in organisms and represent indispensable constituents in intricate biological activities.

Most notably, the majority of RNA in higher organisms is *noncoding RNA (ncRNA*) that lacks protein coding capacity [14]. While *ncRNA* was once considered to be a byproduct of RNA polymerase transcription without any biological function [15], an increasing body of research has demonstrated that *ncRNA* participates in a wide range of intracellular biological processes and plays a critical regulatory role in organismal growth, development, and apoptosis [16,17]. Furthermore, *ncRNA* has been found to be closely associated with a variety of complex human diseases [18,19]. As such, research into the complex and important functions of *ncRNA* has become a crucial component in unraveling the mysteries of life [20]. Regrettably, the instability and diversity of *ncRNA* present significant challenges to the study of its function. However, studies have indicated that *ncRNAs* from the same family exhibit similar functions [21], suggesting that identifying their families can provide preliminary insights into the function of *ncRNAs* and guide further experimental validation of their functions.

Currently, there are two main categories of methods for identifying *ncRNAs*: experimental-based methods and computational-based methods. Each method has its principles, which are discussed below.

### 1.1. Tranditional Experiment-Based Approach

The first experimental-based method involves using chemical or enzyme reagents for *ncRNA* sequencing, where classification and identification are based on the size of *ncRNAs* [22]. This method is relatively simple and independent of *ncRNA* structure, as it does not require reverse transcription of cDNA. The second method involves generating cDNA libraries through reverse transcription to identify *ncRNAs* [23]. This method allows for the creation of specific cDNA libraries tailored to identify particular functional categories of *ncRNAs*. However, the efficiency of reverse transcription can be affected by the structure and modifications of *ncRNAs*, leading to incomplete reverse transcription and the inability to identify all *ncRNAs* from specific families in the library. Base loss during reverse transcription can also impact identification performance [24]. Microarray analysis is the third method used to identify *ncRNAs* by probing their binding [25]. This approach enables the rapid and simultaneous identification of multiple types of *ncRNAs*, even at lower concentrations. It has become a widely used method in transcription detection in research. The fourth method involves using the SELEX technique [26], where *ncRNAs* are identified by forming ribonucleoprotein particles with specific proteins. This technique can generate *ncRNAs* from all genes in an organism, regardless of their abundance in the cell. 

These experimental-based methods share common disadvantages, including complexity, high costs, and limitations in meeting the demands of high throughput *ncRNA* identification.

### 1.2. Machine Learning-Based Approach

Owing to the industry’s pressing need for efficient and expeditious *ncRNA* recognition, computational methods have come to the fore. These computational approaches primarily encompass two principal categories. The first method is based on RNA secondary structure. Infernal is a typical method based on sequence alignment [27]. It first uses secondary structure data to annotate the consistency of *ncRNA* sequences within the same family. Then, it builds covariance models (CM) based on stochastic context-free grammars (SCFGs) using the annotated sequence data. Finally, these covariance models are utilized to accurately identify *ncRNA* families. This type of method sometimes using RNA secondary structure prediction tools such as mfold [28] and Ipknot [29] to predict the secondary structure. Then, algorithms are designed to learn the structural features for *ncRNA* identification. GraPPLE [30], RNAcon [31] and nRC [32] are representative methods in this field. Among them, GraPPLE utilizes global graph features of *ncRNA* secondary structures and designs an SVM method. RNAcon extracts 20 types of secondary structure graph features and employs a random forest method. nRC employs RNA secondary structure and a deep learning model based on convolutional neural networks for recognition. The second approach is based on prediction algorithms for RNA sequence. ncRFP [33] simplifies the process by automatically extracting features from *ncRNA* sequences for predicting *ncRNA* families. Although these models can predict *ncRNAs*, there is still room for improvement in terms of accuracy and other metrics.

The cross multi-head attention mechanism is a method that helps computers focus on different parts of data at once and combine them, which is valuable for complex analysis like understanding intricate relationships and features within RNA sequences in bioinformatics, enhancing the accuracy of analysis. In addition, the Transformer model has gained widespread recognition as a highly influential deep learning algorithm [34]. It has attracted significant attention in the field of natural language processing in recent years. The introduction of the Transformer model has addressed the limitations of conventional Seq2Seq models and has demonstrated remarkable performance in various tasks such as machine translation, text summarization, and dialogue generation. By introducing multi-head self-attention mechanisms, the Transformer model allows for parallelized training, enabling efficient processing of input sequences and capturing the sequential relationships among words, thus improving overall accuracy. This has resulted in rapid expansion of Transformer-based algorithms across diverse domains including computer vision and bioinformatics. For instance, in computer vision, the Visual Transformer (VIT) algorithm has successfully applied the Transformer model to achieve state of the art performance in image classification tasks [35], thereby showcasing the exceptional robustness of the Transformer model. In the field of bioinformatics, attention mechanism-based approaches have also garnered significant attention. The AlphaFold [36] model, as a deep learning-based protein structure predictor, has achieved breakthrough results by leveraging neural network structures like the Transformer. Furthermore, Choi et al. [37] focused on applying the Transformer architecture and attention mechanisms to analyze protein interactions, enriching the methodologies in this domain. Additionally, Cao et al. [38] introduced GGCDA, a novel attention mechanism-based method for predicting associations between circular RNAs and diseases, effectively merging computational methods with biological contexts. These works have also received widespread recognition. Furthermore, Transformer-based algorithms have also demonstrated promising outcomes in tasks such as RNA secondary structure prediction and drug molecule screening and design, showcasing their efficacy in these domains.

This study specifically investigates the potential of utilizing the Transformer model for extracting RNA sequence features within the domains of bioinformatics and drug molecule design. Augmenting the performance of our model in this study entails capitalizing on the inherent capabilities of the attention mechanism and feature compression within the framework of the Transformer model. This study presents a novel deep learning-based approach for classifying noncoding RNA families. The proposed method utilizes a k-mer technique to represent features, thereby enhancing the accuracy of RNA sequence recognition. The RNA sequences are then fed into CNN (convolutional neural network) and BiLSTM (bidirectional long short term memory) models, enabling the extraction of structural and sequential feature relationships within the sequences. To focus on important information and adjust the weights of key details, an MLP module with an integrated attention mechanism is employed to map the features onto a new feature space.

While traditional residual structures exhibit a certain degree of capability in integrating features across different levels of the network, their understanding of the interrelationships among these features remains limited. In our model, the introduction of an attention-based residual neural network architecture facilitates a comparative analysis of RNA features obtained from various levels, yielding relevant correlation information. This approach effectively enhances the fusion of features from heterogeneous network layers, including the structural information identified by the CNN. Specifically, our residual block structure is founded on the intrinsic correlations between shallow and deep network layers, as well as the interdependencies among the features extracted at these levels. As a result, in comparison to conventional methods, our approach excels at preserving information derived from shallower network levels. By incorporating an attention-based residual network architecture, we can better capture features across different scales. This approach also enables the model to effectively capture variations in deep features while maintaining shallow features, thereby enhancing the model’s capacity for data representation and analysis.

The performance of the proposed model is evaluated using ten publicly available datasets. The experimental results demonstrate its significant advantages over alternative algorithms, underscoring its potential in the accurate prediction of *noncoding RNA* families.

## 2. Materials and Methods

In order to investigate the practical performance of the model proposed in this study, publicly available datasets comprising 13 distinct *ncRNAs* were employed as experimental materials. The experimental results were comprehensively compared with those of benchmark algorithms. Drawing upon the unique characteristics of RNA sequences, this study proposes a novel multi-scale residual network model for the prediction of *noncoding RNA* families. The model incorporates bidirectional long short-term memory (BiLSTM), attention mechanisms, and dilated convolutions to capture the inherent complexities of RNA data. To address potential errors in the RNA dataset, a 2-mer approach is employed. Additionally, a feature representation method utilizing word embeddings with an embedding dimension of 16 and a length of 224 is adopted. The BiLSTM and convolutional neural network (CNN) modules are then applied to extract initial features from both the RNA sequence and structure, effectively augmenting the input dimensionality of the model. These extracted features are concatenated to form a sequence of dimensions 224∗128. To facilitate the learning of intricate and abstract representations, nonlinear feature mapping and transformation are achieved through fully connected layers.

Attention mechanisms are employed to compute the disparities between preceding features and those generated by the multi-layer perceptron (MLP), enabling the model to capture abstract information while preserving the original features. Block1, a CNN module encompassing multiple scales, is designed to encompass a convolutional module with a scale of 16, as well as two dilated convolution modules featuring convolution window sizes of 10 and 18, respectively. The integration of attention mechanisms allows for the computation of differences between shallow and deep networks, with the outcomes being added to the shallow network to mitigate overfitting risks associated with excessive network depth. Downsampling is achieved through positional data reshaping, enhancing the thickness of feature representations while maintaining the integrity of the original features. Consequently, the length of the sequence is halved, with the embedding dimension doubled. Block2 inherits the same parameters as Block1 but possesses twice the number of filters. It further extracts global information from the RNA and leverages attention mechanisms to calculate disparities. Ultimately, prediction is performed through fully connected layers and the softmax activation function.

### 2.1. Dataset

The data used in this study were obtained from the Rfam database [39]. Rfam is a comprehensive collection of RNA families, providing a valuable resource for the analysis of *ncRNA* sequences. The database contains 13 distinct types of *ncRNAs*, encompassing *microRNAs*, *5S_rRNA*, *5.8S_rRNA*, *ribozymes*, *CD-box*, *HACA-box*, *scaRNA*, *tRNA*, *Intron_gpI*, *Intron_gpII*, *IRES*, *leader*, and *riboswitch*.

The dataset employed in this study consists of 6320 nonredundant *ncRNA* sequences. This dataset encompasses various RNA families, such as the *IRES* family with 320 sequences, while the remaining families each comprise 500 sequences. To effectively train the model, a ten-fold cross-validation methodology was employed during the model training phase. This approach was selected based on its established effectiveness in validating and assessing the generalization performance of machine learning models. By dividing the data into subsets and iterating through multiple training and testing cycles, this technique provides a comprehensive evaluation of the model’s performance across different data distributions and ensures that the results are robust and reliable. This alignment with a widely recognized validation technique enhances the credibility of our research findings and supports our aim of developing a model that accurately classifies various RNA families (Figure 1).

### 2.2. RNA Representation Method

Better feature representation contributes to more accurate differentiation of RNA families, thereby improving the predictive performance of the model. In this study, we utilized a feature representation method based on k-mer and embedding. Firstly, we selected the k-mer method to process RNA sequences, using 2-mers for RNA feature representation. Subsequently, we applied word embedding techniques to convert the RNA sequences into vector representations based on their frequency, facilitating the processing and training of the network model. The advantages of using k-mer method in RNA and DNA research include:Dimensionality reduction: RNA sequences are often very long, and analyzing the raw sequence data may result in a high dimensional feature space, leading to the curse of dimensionality. Representing RNA sequences as k-mer sequences significantly reduces the dimensionality of the feature space, thus reducing computational complexity and improving processing efficiency.Capturing contextual information: Word embedding maps discrete symbol sequences (such as k-mer) into a continuous vector space, where symbols with similar contexts have similar embedding representations. By converting k-mer sequences into word embedding vectors, we can capture contextual information in RNA sequences, including the associations between nucleotides. This is important for many machine learning and deep learning algorithms, as they can utilize these embedding vectors to infer the functional and structural information of RNA sequences.

### 2.3. Neural Network Architecture

#### 2.3.1. Convolutional Neural Network in ConF

Convolutional neural networks (CNNs) apply convolutional operations to RNA data using convolutional kernels and employ activation functions to introduce nonlinear computations, thus increasing their expressive capacity. The resulting feature maps are then produced as inputs for the subsequent layers. CNNs commonly comprise multiple layers of convolutional layers, where the lower layers mainly extract low level features from the input data, while the higher layers combine these low-level features to extract higher level abstract features. The operation of a convolutional kernel in the i-th layer can be represented by the following equation:(1)$$x_{j}^{l}=f\left(\sum_{i\in M_{j}}x_{i}^{l-1}w_{ij}^{l}+b_{j}^{l}\right)$$

The notation used is as follows: $x_{j}^{l}$ represents the convolutional kernel at position $\left(i,j\right)$ in layer *l*, $x_{i}^{l-1}$ represents the feature map of the $\left(i-1\right)$th layer, $b_{j}^{l}$ represents the bias, and *f* denotes the activation function. The Convolutional kernel, typically smaller than the input data, performs convolutional calculations on a subset of nodes within the input data known as the “receptive field”. This strategy enables the effective extraction of local features from the input data, leading to improved accuracy. Moreover, the convolutional kernel can slide across all positions of the input data, with shared weights during each convolutional operation. This weight sharing mechanism reduces the number of parameters in the network, enhancing the scalability of the network model.

Convolutional kernels have been widely demonstrated as effective feature extraction tools in handling sequence data. Due to the presence of specific local patterns and structures within *ncRNA* sequences, we believe that convolutional kernels excel at capturing these local features, which is crucial for classification tasks. Moreover, convolutional kernels possess learnable parameters that can adaptively capture features of varying scales and patterns, a particularly crucial aspect when dealing with different families of *ncRNA*.

#### 2.3.2. Cross Multi-Head Self-Attention in ConF

The primary function of the cross multi-head attention mechanism (MHA) is to facilitate cross-interaction between two distinct feature sets, enabling each feature to consider information from the other feature set. This mechanism aims to enhance the capture of interactions and correlations between the features. In this paper, the cross multi-head attention mechanism is employed to handle comparisons between different blocks, with each input possessing its unique feature representation. 

Specifically, the cross multi-head attention mechanism enables interaction between features at various levels. At each level, the attention mechanism calculates the similarity between the two feature sets and subsequently computes a weighted sum of elements in each set based on the similarity weights. This process allows each element within each feature set to incorporate information from all elements in the other feature set, resulting in a more effective capture of their associations and interactions. 

The multi-head attention mechanism (MHA) receives three vectors as inputs: the query vector, the key vector, and the value vector. Given a query vector, MHA calculates weighted sums of the key vectors, with the weights determined by the similarity between the query and key vectors. The resulting weighted sum is then multiplied by the value vector to generate the output. Common similarity calculation methods include dot product or bilinear calculations. The multi-head mechanism of MHA significantly enhances the expressive capacity of the model and enables it to learn more diverse and complex features. The formula for multi-head attention is as follows:(2)$$Q_{i}=QW_{i}^{Q},K_{i}=KW_{i}^{K},V_{i}=VW_{i}^{V},i=1,\ldots ,h$$
(3)$$head_{i}=Attention\left(Q_{i},K_{i},V_{i}\right),i=1,\ldots ,h$$
(4)$$MultiHead\left(Q,K,V\right)=Concat\left(head_{1},\ldots ,head_{h}\right)W^{O}$$
where *Q*, *K*, *V* represent the query matrix, key matrix and value matrix respectively, $W_{i}^{Q},W_{i}^{K},W_{i}^{V}$ represent the weight matrices of the query matrix, key matrix and value matrix respectively, $W^{O}$ represents the output weight matrix, *h* represents the number of heads, $head_{i}$ represents the output of the *i*-th head, and “*Concat*” represents the concatenation operation (Figure 2).

In this study, we chose to adopt this type of multi-head attention mechanism. Firstly, this multi-head attention mechanism has demonstrated remarkable effectiveness in the fields of natural language processing and sequence modeling. This mechanism is capable of capturing inherent relationships within sequences while assigning varying weights to information from different positions, thereby better capturing long range dependencies and local patterns within sequences. This is particularly crucial in the context of *ncRNA* family classification tasks, where different families of *ncRNA* might exhibit variations at different positions.

Secondly, the multi-head attention mechanism holds an advantage in enhancing the model’s expressive and modeling capabilities. By simultaneously focusing on information from different scales and aspects, the model can comprehensively understand sequence features, thus enhancing classification accuracy. Moreover, the multi-head mechanism aids the model in retaining richer information across different levels of abstraction, thereby augmenting the model’s understanding and analytical capabilities towards sequences. Furthermore, Transformer’s multi-head attention mechanism also performs effectively on short sequences, especially when strong internal correlations exist within the sequence. This enables our model to maintain consistent performance when dealing with *ncRNA* sequences of varying lengths. In the model, the learning rate, number of iterations, and loss function are depicted as presented in (Table 1).

#### 2.3.3. BiLSTM in ConF

Long short-term memory (LSTM) has proven to be an effective model for handling long range dependencies in sequential data. RNA sequences, being context sensitive data, exhibit a strong correlation between the profile information of each target base and its surrounding context. In this study, LSTM is selected as the fundamental network for extracting target bases and their contextual features and subsequently encoding them. The operation of LSTM begins from one end of the sequence data and progresses to the other end. However, a unidirectional LSTM can only capture information from a single side of the target base. To overcome this limitation and capture contextual information from both sides, this study adopts bidirectional LSTM (BiLSTM) to extract and learn the features of target bases and their corresponding sequence patterns.

BiLSTM is designed to extract and learn features from the input data, facilitating the creation of a model that encodes each base along with its contextual information in a consistent format. BiLSTM is composed of two LSTM networks: a forward LSTM network with 16 hidden nodes that records the contextual features of the target base’s left side, progressing from left to right, and a backward LSTM network with 16 hidden nodes that records the contextual features of the target base’s right side, progressing from right to left. Following the processing stage, the outputs of the two LSTMs are concatenated. The final output of the BiLSTM model can only be obtained when all time steps have been computed. At each base position, BiLSTM generates two hidden states. These two hidden states are then combined at the target base, resulting in the derivation of encoded data (1∗32) representing the target base and its contextual features, which is subsequently outputted.

In the LSTM formula, $f_{t}$ represents the output of the forget gate, which determines which information should be forgotten from the cell state. $i_{t}\;$represents the output of the input gate, which determines which new information should be stored in the cell state. $o_{t}$ represents the output of the output gate, which determines which information in the cell state should be output. The formula for LSTM calculation is as follows:(5)$$f_{t}=\sigma \left(W_{f}\cdot \left[h_{t-1},x_{t}\right]+b_{f}\right)$$
(6)$$i_{t}=\sigma \left(W_{i}\cdot \left[h_{t-1},x_{t}\right]+b_{i}\right)$$
(7)$${\tilde{C}}_{t}=tanh\left(W_{C}\cdot \left[h_{t-1},x_{t}\right]+b_{C}\right)$$
(8)$$C_{t}=f_{t}*C_{t-1}+i_{t}*{\tilde{C}}_{t}$$
(9)$$o_{t}=\sigma \left(W_{o}\cdot \left[h_{t-1},x_{t}\right]+b_{o}\right)$$
(10)$$h_{t}=o_{t}*tanh\left(C_{t}\right)$$

Here, $f_{t}$ is the forget gate, $i_{t}$ is the input gate, ${\tilde{C}}_{t}$ is the new candidate value, $C_{t}$ is the cell state, $o_{t}$ is the output gate, and $h_{t}$ is the hidden state. *σ* is the sigmoid function and tanh is the hyperbolic tangent function. $W_{f}$, $W_{i}$, $W_{c}$, $W_{o}$, $b_{f}$, $b_{i}$, $b_{c}$, and $b_{o}$ are all learnable parameters.

#### 2.3.4. Residual Structure in ConF

In this study, a residual structure was introduced into the proposed model to extract features from the output data learned in the first part and classify them. The incorporation of residual connections allows information to selectively bypass certain layers in the neural network, facilitating the flow of information. The residual structure makes this choice more direct and easier for the network to learn. ResNet, a specialized type of convolutional neural network, employs residual blocks as fundamental units. By utilizing shortcut connections between the input and output layers of the residual block, it combines the input data with the mapped data to generate the output data, ensuring that each residual block in the network incorporates the original input information. This not only improves the model’s trainability but also effectively mitigates the degradation issue that can arise with deeper network architectures. Typically, ResNet comprises a specific number of residual blocks, where the input data are denoted as x, the mapping of the residual block is represented as $F\left(x\right)$, and the output is obtained by the sum of the mapping and the input, i.e., $H\left(x\right)=F\left(x\right)+x$. In ResNet, when adding a new residual block as the network becomes saturated, the mapping function $F\left(x\right)$ can be set to zero, which research has demonstrated to facilitate the implementation of an identity mapping compared to regular convolutional networks.

Capitalizing on the favorable performance and ease of training afforded by residual architectures, this study introduces an innovative paradigm of residual blocks. The blocks of the model incorporate multiple convolutional windows of varying sizes, allowing for the extraction of a broader range of structural features compared to conventional residual network modules. Additionally, the residual component employs a cross multi-head attention mechanism, which, as opposed to the traditional element-wise addition, enables the model to capture feature disparities across different modules more effectively, thereby enhancing the extraction of intrinsic features in RNA.

## 3. Results and Discussion

In order to investigate the practical performance of the model proposed in this study, publicly available datasets comprising 13 distinct *ncRNAs* were employed as experimental materials. The experimental results were comprehensively compared with those of benchmark algorithms, revealing substantial advantages. This chapter provides a visual demonstration of the comprehensive outstanding performance of the proposed algorithm, emphasizing its commendable predictive capabilities across multiple evaluation metrics.

### 3.1. Evaluation Metrics

In order to assess the overall performance of each method across various aspects, this study utilizes accuracy, sensitivity, precision, and F1-score as the evaluation metrics for comparing algorithm performance. The specific calculation methods for accuracy, sensitivity, precision, and F1-score are described below. In this context, TP, TN, FP, and FN represent the counts of true positives, true negatives, false positives, and false negatives, respectively, for the different methods evaluated using the ten-fold cross-validation test set.
(11)$$Accuracy=\frac{TP+TN}{TP+FP+FN+TN}$$
(12)$$Sensitivity=\frac{TP}{TP+FN}$$
(13)$$Precision=\frac{TP}{TP+FP}$$
(14)$$F1-score=\frac{2*TP}{2*TP+FN+TP}$$

### 3.2. Comprehensive Performance Evaluation

The datasets utilized in this study encompassed 13 distinct ncRNA families and served as the basis for comparing the performance of the ConF model with three benchmark models: RNAcon, nRC, and ncRFP. These three algorithms have demonstrated strong performance, utilizing graph-based representation, convolutional neural networks, and attention mechanisms, respectively, for RNA prediction. This selection not only facilitates an in-depth exploration of the impact of different network structures on predictive outcomes but also provides valuable comparative references for the algorithm proposed in this study. Furthermore, the notable transparency of these methods contributes to a solid foundation for subsequent research and comparisons, fostering the continued advancement of the relevant field. Comparative analysis of the experimental results reveals that the proposed ConF model outperforms the three benchmark algorithms in terms of accuracy, sensitivity, precision, and F1-score. Taking a vertical perspective, accuracy and F1-score offer a comprehensive assessment of the model’s performance. The ConF model incorporates a multi-scale CNN module and a residual module with attention mechanisms, which enables the algorithm to better extract the intrinsic features of RNA compared to other benchmark algorithms. Regarding accuracy, the ConF model demonstrates superiority of 0.5831, 0.2608, and 0.1596 over the other three models, respectively. In terms of F1-score, the ConF model exhibits a superiority of 0.6051, 0.2678, and 0.1673 over the other three models, respectively. Based on these findings, it is evident that the ConF model exhibits a substantial superiority over the benchmark algorithms in terms of accuracy and F1-score, thereby highlighting its exceptional overall performance. Notably, the ConF model also exhibits noticeable advantages in sensitivity and precision compared to other algorithms, suggesting its capability to detect a greater number of ncRNA families and accurately filter out irrelevant RNA sequences, thereby enhancing prediction precision. The results indicate that the ncRFP model, based on the attention mechanism, outperforms the nRC model employing convolutional methods in terms of predictive capability. Furthermore, both the ncRFP and nRC models exhibit superior predictive performance compared to the RNAcon graph-based approach. This comparison underscores the effectiveness of attention- and convolution-based methods in enhancing predictive accuracy for the current task. The ncRFP model leverages attention mechanisms to capture intricate intrinsic relationships within the data, resulting in exceptional performance, while the nRC model excels through its convolutional approach. In contrast, the RNAcon graph-based model may have limitations in capturing complex patterns within the data, leading to comparatively lower predictive performance. Importantly, our approach introduces techniques such as multiple convolutions to further enhance network performance, thereby substantiating the efficacy of our algorithmic framework.

Taking a horizontal perspective, the ConF model attains the highest performance in terms of accuracy and precision, reaching an exceptionally high value of 0.9568. Additionally, it achieves the best performance in terms of F1-score and sensitivity, surpassing the benchmark algorithms with values of 0.9556 and 0.9553, respectively (Table 2).

### 3.3. Performance Comparison of Diferent Families

Through meticulous data analysis, we have gleaned the following observations and analysis regarding model performance:

In terms of sensitivity, the “ncRFP” and “ConF” models consistently exhibit pronounced advantages across a majority of RNA families. This signifies their heightened sensitivity in detecting positive class samples, which is pivotal for accurate RNA family classification, as capturing positive class features accurately plays a crucial role. The F1-score, encompassing both precision and recall, provides a comprehensive metric for evaluating performance. Notably, in the majority of cases, both the “ncRFP” and “ConF” models achieve high F1-scores, indicating their significant competitive edge in balancing classification accuracy and comprehensiveness. Precision accentuates the accuracy of positive class predictions. Of particular note is the higher precision demonstrated by the “ncRFP” and “ConF” models across multiple RNA families, underscoring their outstanding performance in correctly predicting positive class samples. This holds pivotal importance in avoiding false predictions and accurately identifying crucial RNA families.

In summary, the “ConF” model consistently demonstrates notable superiority across various metrics in the majority of instances. This could be attributed to its effective integration of the cross multi-head attention mechanism within its design, enabling it to better capture critical information within RNA sequences (Figure 3, Figure 4 and Figure 5).

### 3.4. Performance Testing Based on Different Embedding Methods

The choice of RNA representation holds a pivotal position in maintaining its intrinsic characteristics, thereby exerting a profound influence on the effectiveness of RNA category prediction models. Empirical results underscore the divergent outcomes stemming from the manipulation of k-mer method lengths. In our investigation, we incorporated single, 2-mer, and 3-mer sequence segmentation techniques for comprehensive evaluation. Impressively, the model demonstrated the most superior mean accuracy with k = 2, surpassing the accuracy rates of k = 1 and k = 3 by 0.265% and 0.314%, respectively. This illuminates the significance of selecting an optimal k-mer length to enhance the model’s predictive capabilities and underscores the nuanced interplay between feature extraction and performance in RNA classification (Figure 6). 

### 3.5. Correlation Analysis

The correlation matrix in Figure 7 illustrates the relationships between F1-scores of 13 *ncRNA* types predicted by the ConF algorithm. Each cell in the matrix represents the correlation coefficient between the predicted F1-score of an *ncRNA* and its corresponding *ncRNA* category. Higher values closer to 1 indicate a stronger positive correlation, while values closer to −1 suggest a stronger negative correlation (Figure 7).

For example, the correlation coefficient of 0.54 between *5S_rRNA* and *tRNA* indicates a positive correlation, implying that these two RNA categories exhibit similar features that allow the model to make positively correlated predictions. Conversely, the correlation coefficient of −0.34 between *5S_rRNA* and *ribozyme* suggests a weak correlation, indicating that the model struggles to extract relevant features distinguishing these two *ncRNA* categories.

Moreover, there are variations in the correlation of F1-score prediction values across different *ncRNA* categories. Comparing *5S_rRNA* with *Intron_gpI* shows a relatively low correlation, whereas the correlation between *5S_rRNA* and *tRNA* is high. This discrepancy suggests that the correlation between the same *ncRNA* type and different *ncRNA* types likely varies, possibly due to the model’s bias in feature extraction and the inherent differences in *ncRNA* characteristics. These correlation coefficients could potentially be used to study feature similarity and functional similarity between RNA categories.

### 3.6. Robustness Analysis

In order to comprehensively evaluate the robustness of the proposed model, this study adopted a rigorous methodology. The dataset was partitioned into five distinct subsets, each subjected to a ten-fold cross-validation process resulting in the generation of five distinct distributions of model accuracy. These distributions were effectively visualized using box plots, facilitating an intuitive observation of the dispersion of model performance across diverse testing conditions, thereby shedding light on its stability (Figure 8).

Within the array of five box plots generated, the representation of model accuracy distribution for each individual subset becomes evident. A meticulous examination of key statistical parameters within these box plots, such as upper and lower quartiles, median, and outliers, allowed for a meticulous assessment of the model’s performance during each testing iteration. A narrower range of distribution and a stable median within the box plot would indicate minimal fluctuations in accuracy across diverse tests, thereby indicating a higher level of stability. Conversely, a broader range of distribution and significant variation in median values would imply noticeable accuracy fluctuations among different tests, revealing a lower level of stability.

Through an extensive analysis of the outcomes derived from the five box plots, more refined and nuanced conclusions concerning the model’s robustness could be drawn. This visual approach grounded in the utilization of box plots provides a holistic comprehension of the performance fluctuations exhibited by the model, effectively offering robust evidence to further underpin the evaluation of the model’s reliability.

### 3.7. Relationship between Iterations and Performance

Based on the given data, we conducted an analysis on the relationship between the number of iterations and accuracy. It is evident that as the number of iterations increases, the accuracy initially exhibits a rising trend followed by a subsequent decline. During the initial iterations, there is a rapid growth in accuracy, surpassing 90% by the 28th iteration. Subsequently, the rate of accuracy improvement slows down, accompanied by a deceleration in loss reduction, while still maintaining an overall upward trend. However, beyond a certain number of iterations, a slight decrease in accuracy is observed, although the overall trend remains positive, indicating a continued increase in accuracy (Figure 9).

## 4. Conclusions

Given the inherent limitations in *ncRNA* recognition, there is an urgent need to develop computationally efficient methods. This study introduces an innovative deep learning-based approach, named the ConF model, for predicting the classification of noncoding RNA families. The ConF model is specifically designed to address the performance and applicability constraints of current algorithms, leveraging techniques such as attention mechanisms and convolutional methods to effectively extract feature information from *ncRNA* sequences, resulting in a significant enhancement in prediction accuracy. A notable advantage of the ConF model is its exclusive reliance on sequence data, enabling broad applicability even with limited data availability. Experimental results demonstrate a significant performance improvement compared to several benchmark methods, positioning the ConF algorithm not only as a promising solution for predicting RBP binding sites but also as a potential support for functional and medical research related to noncoding RNA. In the context of future research directions, we will consider employing more reasonable residual modules or enhancing the model’s performance by increasing its depth and adjusting convolutional kernels more appropriately. Additionally, the exploration of using other gene sequence prediction data for validation could enhance the model’s versatility and robustness. These explorations will contribute to a deeper understanding of this field and provide valuable guidance for future research endeavors. 

## Figures and Tables

**Figure 1 biomolecules-13-01643-f001:**
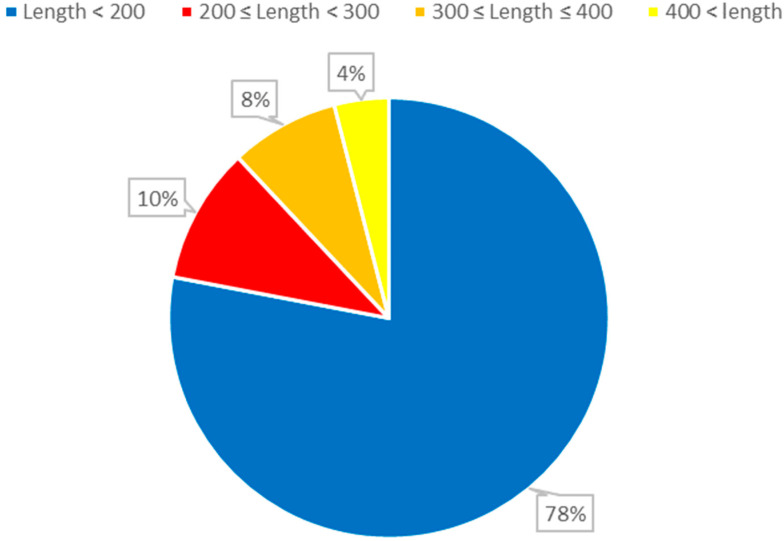
The distribution of ncRNA sequence lengths. The length of RNA refers to the number of nucleotides (AUCG) present in the RNA sequence.

**Figure 2 biomolecules-13-01643-f002:**
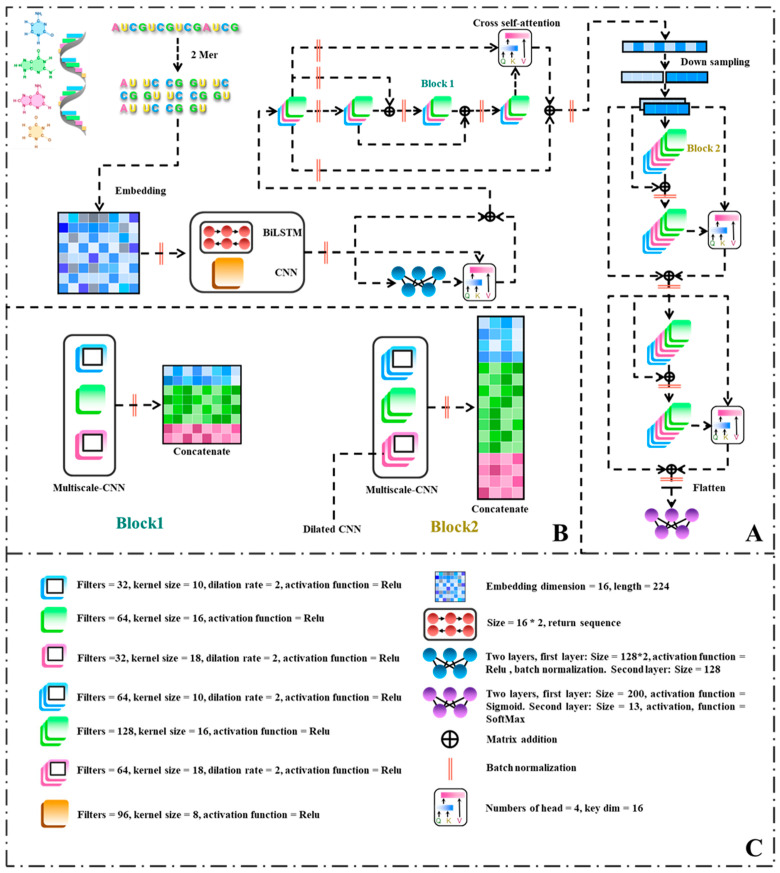
Flowchart of the ConF algorithm: (**A**) Overall architecture of the algorithm; (**B**) Internal structures of Block 1 and Block 2; (**C**) Parameters used in each module of the algorithm along with their explanations.

**Figure 3 biomolecules-13-01643-f003:**
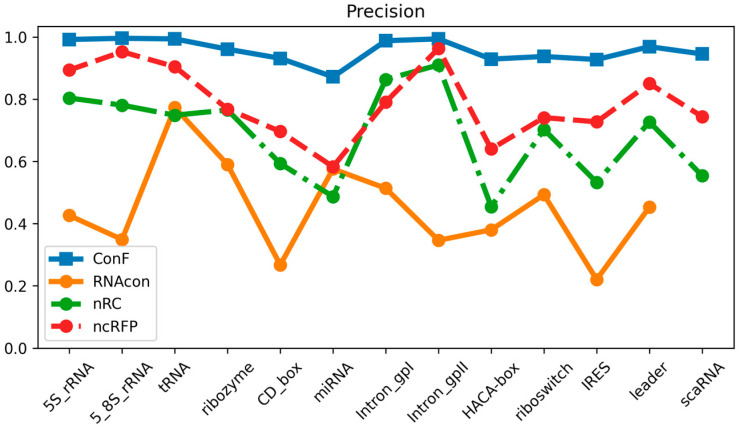
This is a performance comparison among different families. The blue curve, yellow curve, green curve, and red curve represent the performance of ConF, RNAcon, nRC, and ncRFP, respectively.

**Figure 4 biomolecules-13-01643-f004:**
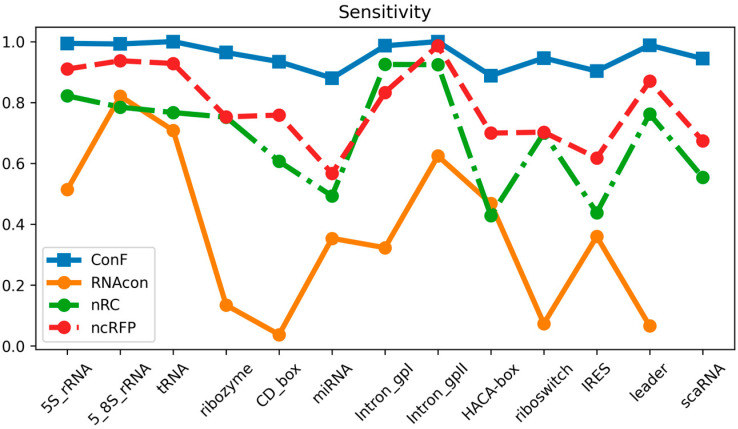
This is a performance comparison among different families. The blue curve, yellow curve, green curve, and red curve represent the performance of ConF, RNAcon, nRC, and ncRFP, respectively.

**Figure 5 biomolecules-13-01643-f005:**
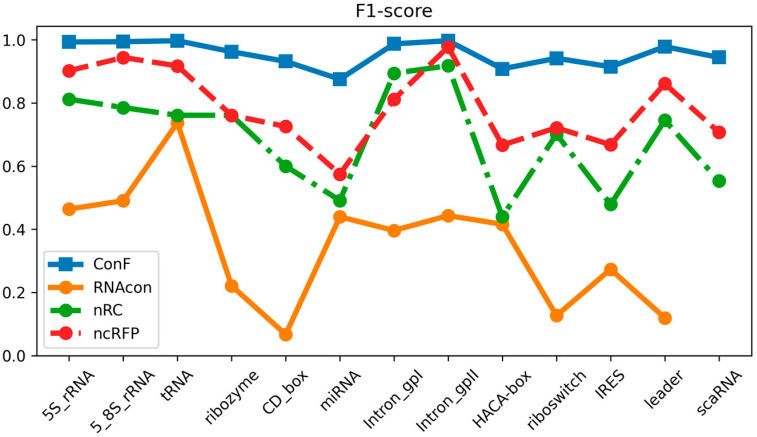
This is a performance comparison among different families. The blue curve, yellow curve, green curve, and red curve represent the performance of ConF, RNAcon, nRC, and ncRFP, respectively.

**Figure 6 biomolecules-13-01643-f006:**
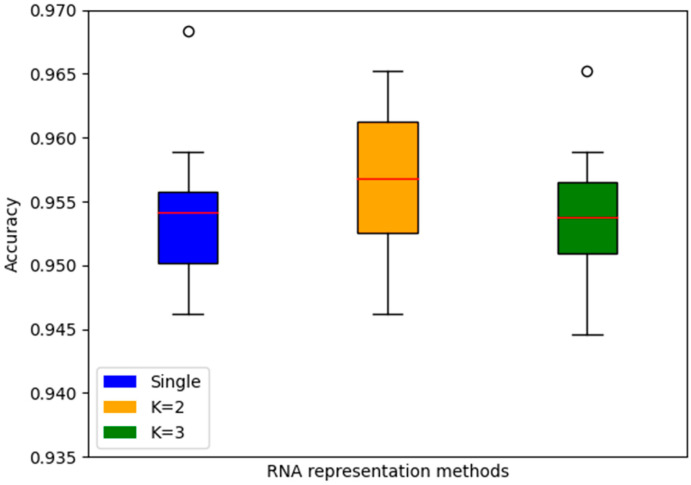
Performance comparison of different encoding methods. The three boxplots in the figure represent the accuracy distributions of three feature representation methods in a ten-fold cross validation. The blue box, orange box, and gray box represent the accuracies obtained using the independent segmentation, 2-mer, and 3-mer methods for RNA sequence representation, respectively.

**Figure 7 biomolecules-13-01643-f007:**
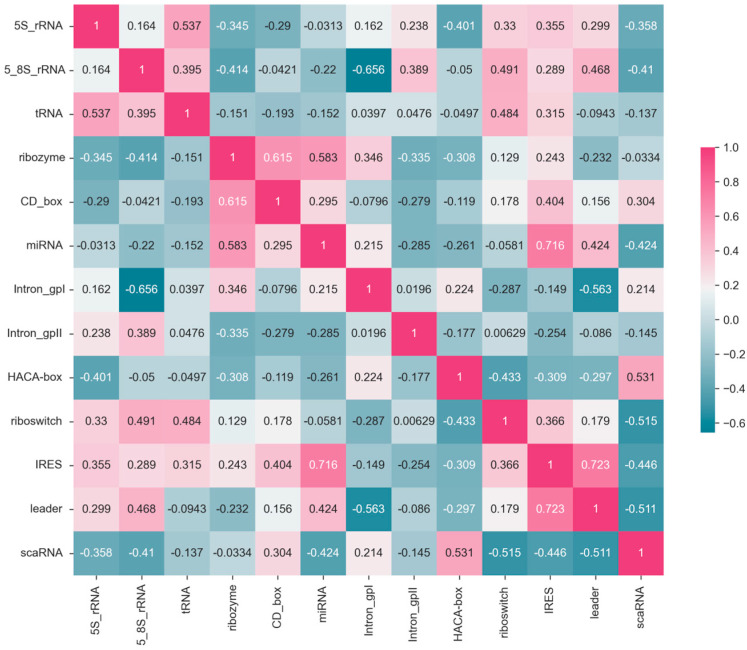
Based on the ConF algorithm, a correlation matrix of precision for each RNA family in a ten-fold test. Each cell in the matrix represents the correlation of classification F1-score between two RNA families. The correlation ranges from 1 to −0.6, where cells closer to magenta indicate stronger correlation, while cells closer to navy blue indicate weaker correlation.

**Figure 8 biomolecules-13-01643-f008:**
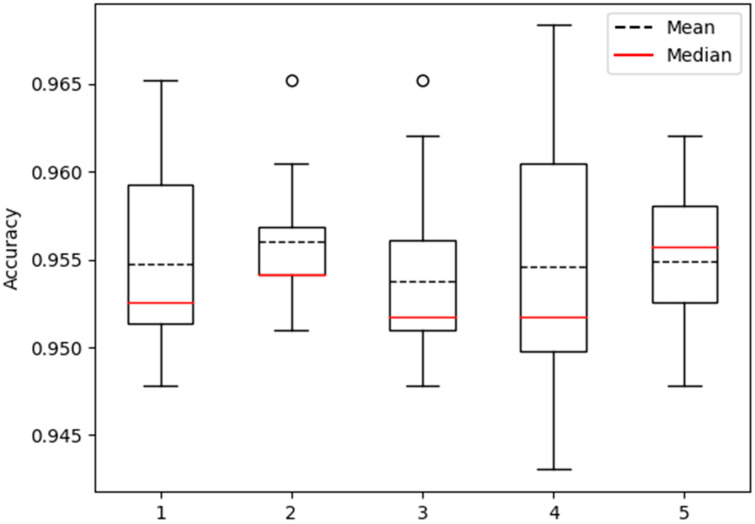
Robustness testing based on five rounds of ten-fold cross-validation.

**Figure 9 biomolecules-13-01643-f009:**
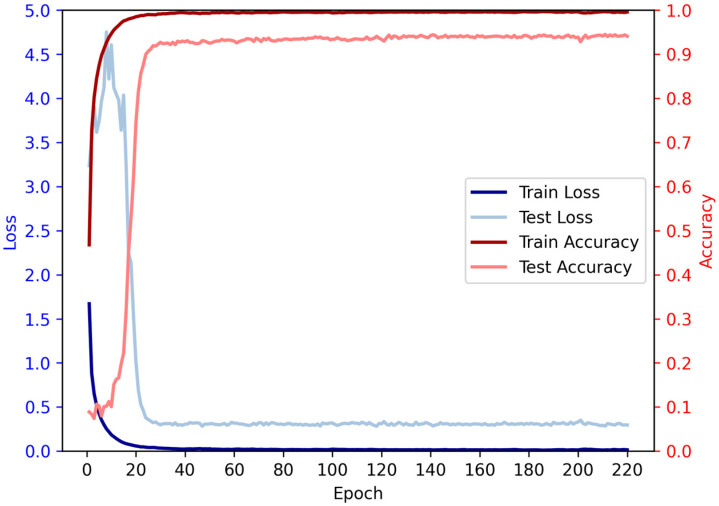
The average accuracy and loss of each epoch in ten-fold cross-validation.

**Table 1 biomolecules-13-01643-t001:** Other parameter settings.

Learning Rate	Epoch	Batch Size	Loss Function
0.001	220	200	categorical Cross entropy

**Table 2 biomolecules-13-01643-t002:** Performance comparison of each method.

Model	Accuracy	Sensitivity	Precision	F1-Score
ConF	**0.9568**	**0.9553**	**0.9568**	**0.9556**
RNAcon	0.3737	0.3732	0.4497	0.3505
nRC	0.6960	0.6889	0.6878	0.6878
ncRFP	0.7972	0.7878	0.7904	0.7883

The best value in each column is bolded.

## Data Availability

Data availability statements are available at Appendix A.

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
