# Peer review of "ConF: A Deep Learning Model Based on BiLSTM, CNN, and Cross Multi-Head Attention Mechanism for Noncoding RNA Family Prediction"

_biomolecules, 2023, doi:10.3390/biom13111643_

Round 1

Reviewer 1 Report

Teragawa and Wang proposed a deep learning method to predict non-coding RNA (ncRNA) families. ncRNA plays important roles in biological processes and the accurate prediction helps the understanding of how ncRNA regulate cellular functions. They thus proposed a computational method to complement the experimental methods. Their method integrates multiple deep learning frameworks, such as BiLSTM, CNN, transformers, residual net and so on. They described the detail of each framework, and then evaluated their method on 13 ncRNA families with 6,320 non-redundant ncRNA sequences. With a single test of 10-fold cross-validation, they claimed that their method achieves superior performance to three existing methods. The topic in the manuscript is interesting, and it is great to show how deep learning can be used in this type of research, but this research needs substantial improvement.

1. No detail about how to train the models and what hyperparameters are used. Also, it is not clear how to use the existing models to make predictions on the dataset. Although each framework is described in detail, I have no idea how to integrate the output of all bases from a ncRNA sequence for a single output prediction.

2. It is wonderful to use 10-fold cross-validation. But a single run of 10-fold cross-validation does not make sense as a main result. The authors need to run 10-fold cross-validation at least 5 times ( 5*10=50 tests) with random data splitting each time to generate robust evaluation.

3. I am glad to see the shared github. However, only dataset is shared, and no code and reproducibility documents are shared to re-generate their performance. As a paper to demonstrate the performance of a new method, code sharing is a must.

4. Figures 3-5 are used twice for different figures. For all figures with line plots, it is better to use line types (dotted, dashed …) together with colors to help those readers without a color printer.

5. It is not clear how to calculate the correction in section 3.5. Also, in line 408, it states that it is about F1-score, but later in line 430, it says that it is about prediction. Which one is true?

6. Lines 375-381 is just an identical copy of text in lines 345-351

7. Many typos are present in the paper. For example, `-` is used random sometimes, incomplete sentence in line 158, and typos in 303, 369, and so on.

There are many typos in the paper and needs a proofreading. 

Author Response

First and foremost, we would like to express our gratitude for the invaluable suggestions you provided for our paper. Your professional insights and review have helped us enhance the paper, making it more comprehensive and credible. This has significantly contributed to the quality of our research work and academic contributions. Once again, we sincerely appreciate your contributions and patience, as your feedback has been highly valuable to us.

Reviewer 2 Report

Firstly, I would like to commend the authors for their diligent effort in developing an intriguing study that explores the attention mechanism in RNA feature extraction. The proposed model, with its integration of convolutional calculations and the Cross Multi-Head Attention mechanism, presents a novel approach. However, several aspects of the paper require further elucidation, refinement, or justification to substantiate the findings. My comments are delineated below.

Introduction:

  • The introduction could benefit from a more robust problem statement or research gap that the study aims to address. The motivation behind the exploration of the attention mechanism in RNA feature extraction should be more explicitly articulated.
  • While the manuscript delves into the complexities of ncRNA recognition, there is a lack of references to substantiate the reported challenges and limitations of existing methods. Please ensure that each statement is adequately supported with scholarly references.
  • A discussion on why the specific attention mechanism was chosen for this study and what makes this approach unique or different compared to traditional methods would enhance the reader's understanding.
  • Lines 93-95: When introducing machine learning-based approaches, it would be beneficial to provide an explanation for non-experts on what the Cross Multi-Head Attention mechanism is, and why it is suited to this kind of analysis.
  • Incorporate analysis of existing methods, including the following referenced papers into the Related Work section: (DOI: https://doi.org/10.3390/biology12071033) (DOI: https://doi.org/10.3390/biom12070932)

Methods:

  • Lines 235-273: The explanation of the convolutional kernel and the Cross Multi-Head Attention mechanism is clear, but it would be beneficial if you could discuss more on why you decided to use this specific approach instead of other available methods. Why is this particular mechanism the most suitable for your research?
  • Lines 192-200: The ten-fold cross-validation methodology employed during the model training phase is well-described. However, kindly provide a solid justification for these choices. Why have you chosen this specific validation technique, and how does it align with the objectives of your research?
  • Lines 253-268: You have presented the formula for Multi-Head Attention, but it would be helpful if you could provide more detailed reasoning or references to justify why this specific formula was chosen and how it contributes to the overall effectiveness of the model.

Results and Discussion:

  • Lines 336-371: The results section provides a comprehensive comparison with benchmark algorithms, revealing substantial advantages. However, a more detailed explanation of the reasoning behind the selection of these specific benchmark models would be helpful. Additionally, a discussion of how the proposed model compares to other models used for similar purposes would provide context.
  • Lines 358-361: The superiority of the proposed model over other models is mentioned, but the manuscript could benefit from a more in-depth analysis of why the model performs better. What are the fundamental differences that lead to this superior performance?

Conclusion:

  • The conclusion should summarize the key findings and contributions of the study, highlighting the novelty and significance of the proposed model in the context of RNA feature extraction.
  • It would be valuable to discuss potential future directions for this research, including possible improvements, extensions, or applications of the proposed model.

Author Response

First and foremost, we would like to extend our heartfelt appreciation for your exceptional review. The professional and invaluable suggestions you've provided have greatly enhanced the quality and depth of our paper. Your review work is not only crucial for the refinement of our current research but also offers valuable guidance for our future research directions. We are particularly thankful for your patience and expertise; your insights have undoubtedly played a significant role in our research and academic growth. Once again, we sincerely thank you for your dedication and outstanding contributions. Your review is an integral part of our research.

Round 2

Reviewer 1 Report

Table 1: “Batch size” appears twice.

Codes are NOT shared yet. As a paper to demonstrate the performance of new methods, shared codes with reproducibility documents must be shared during review process. Otherwise, a manuscript will be rejected.

Author Response

I have made the following revisions to the paper as per the provided request:

1.Removed redundant content from Table 1.
2.Uploaded the complete code and data. In the code section, I have added comments to enhance code readability. This compressed package contains both the dataset and the code, and it should be placed in the appropriate directory to run this program (The experimental setup for this paper utilizes a Lenovo R9000K with an RTX 3080 version and performance mode enabled.).
3.We will make the materials publicly available on GitHub, along with personal contact information, three days after receiving the notification, in order to facilitate readers' understanding of the paper.

Reviewer 2 Report

The authors have sufficiently addressed my concerns. With these revisions, I recommend that the manuscript be published in its current form.

Author Response

Thank you, and best wishes.

Round 3

Reviewer 1 Report

I spent time reviewing the manuscript again. Please kindly write a simple document of how to reproduce the results and share the code in the github repository. I have no other questions.

NO

Author Response

We have already uploaded the complete code and data to 'https://github.com/FROZEN160/RNA-Family,' which includes model explanations and usage instructions. We have optimized the code to make it easier to implement.

Round 4

Reviewer 1 Report

The code was updated two days ago. I have no other questions. 

no